

# The application of blockchain technology in data trading: a systematic review

Wei Xiong[1,2,3], Huaibin Shao[2] and Hong Ge[1]

[1] School of Business Administration, Nanchang Institute of Technology, Nanchang, Jiangxi, China
[2] Jiangxi University of Software Professional Technology, Nanchang, Jiangxi, China
[3] Aheadsoft Software Co., Ltd., Nanchang, Jiangxi, China

## ABSTRACT

In the era of exponential data growth, the imperative to establish secure and efficient data trading mechanisms has become paramount. While traditional centralized architectures present critical limitations in security and operational efficiency, the emergence of blockchain technology offers transformative potential for decentralized solutions. This study conducts a systematic literature review to critically examine blockchain's evolving role in data trading ecosystems. Adhering to the PRISMA 2020 framework, we analyzed 18 rigorously selected studies from an initial pool of 164 Web of Science publications identified through "data trading" and "blockchain" keyword searches. Our analysis reveals three principal findings: first, current blockchain implementations predominantly cluster within computer science applications, indicating disciplinary concentration. Second, technical development emphasizes solution-oriented systems over theoretical model construction, suggesting an application-prioritized research paradigm. Third, we identify persistent challenges across three critical dimensions: (i) security-efficiency paradox in decentralized architectures, (ii) transparency-privacy equilibrium maintenance, and (iii) scalability constraints under high-concurrency scenarios. This study aims to offer in-depth insights into blockchain's potential applications in data trading and future research directions.

## INTRODUCTION

With the rapid development of cloud computing and Internet-of-Things technologies, the volume of global data generation is growing exponentially, and human society has fully entered the big data era (*Botta et al., 2016*). Against this backdrop, data sharing and trading are becoming an inevitable trend for the development of the digital economy, and their market potential has been widely concerned by the academic and industrial circles (*Sutherland & Jarrahi, 2018*). The value of data as a new type of production factor is being deeply explored by various industries (*Lee, 2017*), and the construction of data trading platforms has become an important way to achieve data circulation, giving rise to diversified models such as autonomous data sales, data processing and resale, and platform-based services (*Liang et al., 2018*; *Peters, 2023*).

The current mainstream data trading platform service models are mainly divided into two categories: the custody-based trading model and the aggregation-based trading model. In the custody-based model, institutions fully transfer their data assets to a centralized

Corresponding author
Huaibin Shao,
huaibinshao16@163.com

database. The platform has complete control over the subsequent data trading and application, and the original data providers are completely separated from the trading process (*Wang et al., 2018*; *Liu & Lin, 2023*). For example, the choice of trading frequency and trading partners depends entirely on the platform's unilateral decision-making, and the protection of data rights essentially relies on the platform's credit endorsement. In contrast, the aggregation-based model dynamically connects multiple data sources through application programming interfaces. Business institutions retain data management rights and only interact in real-time *via* the platform when a query request is triggered (*Xiong & Xiong, 2019*). Although this model appears to maintain distributed control of data, an in-depth analysis of the data flow mechanism reveals that the platform can still gradually gain control over data by accumulating transaction logs, eventually turning into a *de facto* custody-based model (*Yu et al., 2020*).

However, the centralized management mechanism mentioned above is becoming a significant barrier to data circulation, causing multiple systemic risks. First, the lack of trust among trading parties leads to fairness disputes, especially in complex trading scenarios with multiple participants (*Chen et al., 2019a*). Second, centralized platforms' data retention practices increase the risks of privacy breaches and data tampering (*Zhao et al., 2019*; *Yang et al., 2020*). Third, single points of failure and low transparency severely affect the availability and credibility of transactions (*He et al., 2019*; *Hu et al., 2021*). Recent research further indicates that insufficient data availability and unclear rights ownership have become key bottlenecks hindering the market's large-scale development (*Yu et al., 2024*; *Bauer-Hänsel et al., 2024*). It is clear that relying solely on platform promises cannot create a trustworthy trading environment. There is an urgent need to rebuild the data rights protection system through technical mechanisms (*Li et al., 2018*; *Janssen et al., 2020*).

To address these challenges, decentralized trading models based on blockchain and smart contracts have become a research focus. Blockchain technology, following the steam engine, electricity, and the internet, is regarded as the next-generation disruptive technology and a world-class computing paradigm. It represents a revolutionary technological advancement, serving as a completely new application model of computer technologies such as distributed data storage, peer-to-peer transmission, consensus mechanisms, and encryption algorithms. Characterized by decentralization, immutability, anonymity, openness, and self-governance (*Nakamoto, 2008*), blockchain technology primarily addresses the safety and fairness issues of transactions in trustless environments (*Pasdar, Lee & Dong, 2023*). A smart contract, defined by *Szabo (1997)* as a computer protocol designed to propagate, validate, or execute contracts, enables traceable and irreversible transactions without the need for a trusted third party. These contracts are marked by transparency, trustworthiness, automatic execution, and enforcement, and are used to ensure the automatic and faithful execution of transactions in trustless environments (*Xiong & Xiong, 2021*). The integration of blockchain and smart contracts offers core advantages for data trading, including decentralization, anonymity, and non-repudiation (*Puthal et al., 2018*; *Egunjobi et al., 2024*). For instance, public key hash

addresses provide anonymity for trading parties (*Allouche et al., 2021*), while dynamic ownership tracking technology strengthens the rights of data owners (*Fadler & Legner, 2022*). On-chain encrypted storage combined with real-time verification mechanisms effectively safeguards privacy and ensures transaction non-repudiation (*Cai et al., 2024*; *Oh et al., 2020*). In theory, this model promises to systematically resolve issues like lack of fairness, privacy breaches, and single points of failure prevalent in traditional centralized platforms. However, the actual effectiveness of this approach still requires rigorous verification through comprehensive research.

This article aims to clarify the research progress and technical path of decentralized data trading models supported by blockchain technology, providing a theoretical basis for solving the structural defects of the existing trading system. The structure of this article is as follows: "Literature Review" reviews the research *status quo* of data trading models and blockchain applications; "Methodology" expounds the research methodology; "Exploratory Analysis of Results" presents the exploratory analysis results; "Limitations and Discussions" discusses the theoretical and practical significance of the research findings and explores the limitations and trade-offs of blockchain in data trading applications; "Conclusions" concludes the whole article and looks forward to future research directions.

## LITERATURE REVIEW

As a core mechanism for the circulation of production factors, data trading has formed a multi-dimensional research system. Early review studies mainly focused on traditional transaction frameworks: *Liang et al. (2018)* deconstructed price-formation mechanisms, trading-process designs, and data-protection strategies, yet their analysis was confined to centralized market paradigms, failing to predict the disruptive impact of decentralized technologies on trading structures. *Xiong & Tang (2021)* combed through data-entitlement and pricing research from economic, legal, and computer-science perspectives. Although they revealed the complexity of entitlement definition, they didn't explore technology-enabled dynamic entitlement paths in depth. Literature on pricing strategies (*Cai et al., 2021*; *Zhang, Jiang & Yuan, 2021*; *Zhang, Beltrán & Liu, 2023*) built theoretical frameworks of game-theory and auction models, but generally lacked technical solutions to the "multiple selling of one item" risk caused by data replicability. Subsequent studies (*Fu & Wang, 2022*; *Wang, Xia & Pei, 2023*) expanded to market-operation mechanisms and data-factor connotations but were still limited by the centralized-platform assumption, finding it hard to break the centralized data-sovereignty dilemma.

In recent years, the integrated application of blockchain technology has offered a new paradigm to overcome the aforementioned limitations. Early studies (*Guan, Shao & Wan, 2018*; *Yang et al., 2020*) have confirmed the fundamental value of its decentralized, traceable, and tamper-proof features in data storage and transaction auditing. The latest advances show three major breakthroughs:

1. Dynamic entitlement and dispute resolution: automatically execute ownership transfer *via* smart contracts (*Ali, Norman & Azzuhri, 2023*; *Liu et al., 2024*), verify ownership

using zero-knowledge proof (*Zhou et al., 2023*; *Xiong et al., 2024*; *Li et al., 2024a*), and effectively curb the risk of data asset loss (*Driessen, Monsieur & Van Den Heuvel, 2022*; *Chen et al., 2024a*; *Feng et al., 2024*).

2. Full-dataset availability guarantee: based on sharding technology and cross-chain protocols, achieve integrity verification and secure aggregation of fragmented data (*Song et al., 2023*; *Abla et al., 2024*), and break through the partial data supply bottleneck of traditional models (*Wang, 2017*; *Chen et al., 2024b*; *Li et al., 2025*).

3. Non-repudiable transaction architecture: using Merkle trees and timestamped chain storage, build a real-time verification mechanism for multiple parties (*Fernandez, Subramaniam & Franklin, 2020*; *Fang et al., 2024*; *Deng, Zuo & Li, 2025*), and significantly enhance transaction transparency and traceability.

However, existing research still has significant limitations. First, most blockchain solutions rely on high-computational-power consensus mechanisms, making them ill-suited for high-frequency real-time transactions (*Jiang et al., 2023*). Second, there is no universal framework for balancing privacy protection and data availability, as performance bottlenecks of technologies like homomorphic encryption hinder practical deployment (*Yang et al., 2024*). Third, cross-jurisdictional data compliance remains unaddressed (*Cai et al., 2024*). These gaps indicate that data trading-model innovation needs to integrate technological breakthroughs and institutional design, not just rely on technical tools.

## METHODOLOGY

This study aims to systematically analyze the current applications of blockchain technology in data trading, with a focus on evaluating its role in ensuring security and fairness throughout transaction processes. By synthesizing high-quality, peer-reviewed literature (2018–2024), we seek to address key challenges and provide actionable insights for improving trust in decentralized data trading. We employed a rigorous three-stage process combining randomized controlled process, blockchain-specific keyword filtering, and quality-assessment tools to ensure methodological robustness. The analysis was guided by the PRISMA 2020 framework and conducted using Web of Science's Core Collection database (accessed October 2024) to maintain academic rigor.

### Tools and frameworks

1. PRISMA 2020 Framework: structuring the systematic review process for transparency and reproducibility, documenting every stage from database-search to result synthesis.
2. AMSTAR 2: evaluating the methodological quality of included studies, focusing on abstract-based criteria such as protocol registration, validity of inclusion criteria, and risk-of-bias assessment.
3. Randomized controlled process: applying *Del Mar & Hoffmann (2015)* methodology to minimize selection bias by randomly prioritizing search terms during the topic selection phase.

## Data collection process

1. Database selection: Web of Science Core Collection was chosen for its curation of high-impact journals, reducing noise from low-quality sources.
2. Initial search: a title-based search using keywords "data trading" and "blockchain" yielded 164 articles.
3. Filtering workflow:

   (1) Stage 1: removed duplicates and non-peer-reviewed articles ($n = 0$ for both).
   (2) Stage 2: excluded irrelevant topics ($n = 71$) *via* keyword analysis, including 69 articles deemed outside the scope of decentralized security applications after title/abstract screening, and two unretrieved reports.
   (3) Stage 3: screening the remaining 93 articles *via* AMSTAR 2 retained only those meeting all seven critical domains (*Shea et al., 2017*), reducing the pool to 18 high-quality studies (see Fig. 1 for the PRISMA flowchart).

## Quality assurance measures

(1) Dual independent reviewers: two researchers conducted title/abstract screening and full-text analysis to mitigate bias, with discrepancies resolved *via* third-party arbitration.
(2) Term precision: explicitly excluded terms like "smart contract optimization" to avoid scope creep, ensuring alignment with the study's security/fairness focus.

## Rationale for methodology

The randomized controlled process minimized researcher bias, while PRISMA's transparent reporting enhanced reproducibility. AMSTAR 2's emphasis on protocol registration and risk-of-bias evaluation ensured that only rigorously designed studies informed our conclusions. This multi-stage approach balances comprehensiveness with precision, addressing gaps identified in prior fragmented reviews.

This structured methodology enabled us to derive evidence-based insights into blockchain's technical and governance contributions to secure equitable data trading ecosystems.

## EXPLORATORY ANALYSIS OF RESULTS

Systematic literature review, as proposed by *Xiao & Watson (2019)*, is a research methodology designed for the exploratory analysis of previously reported results. This approach is commonly used to assess researchers' interest in specific topics (*Snyder, 2019*). It involves collecting and reanalyzing existing research results from both primary and secondary sources.

Table 1 summarises the studies further analysed in this review, listing the authors and main elements of each. Based on this, we classified the reviewed studies into several groups: research on secure data trading as 'Security', on fair data trading as 'Fairness', and on both secure and fair data trading as 'Security & Fairness'.

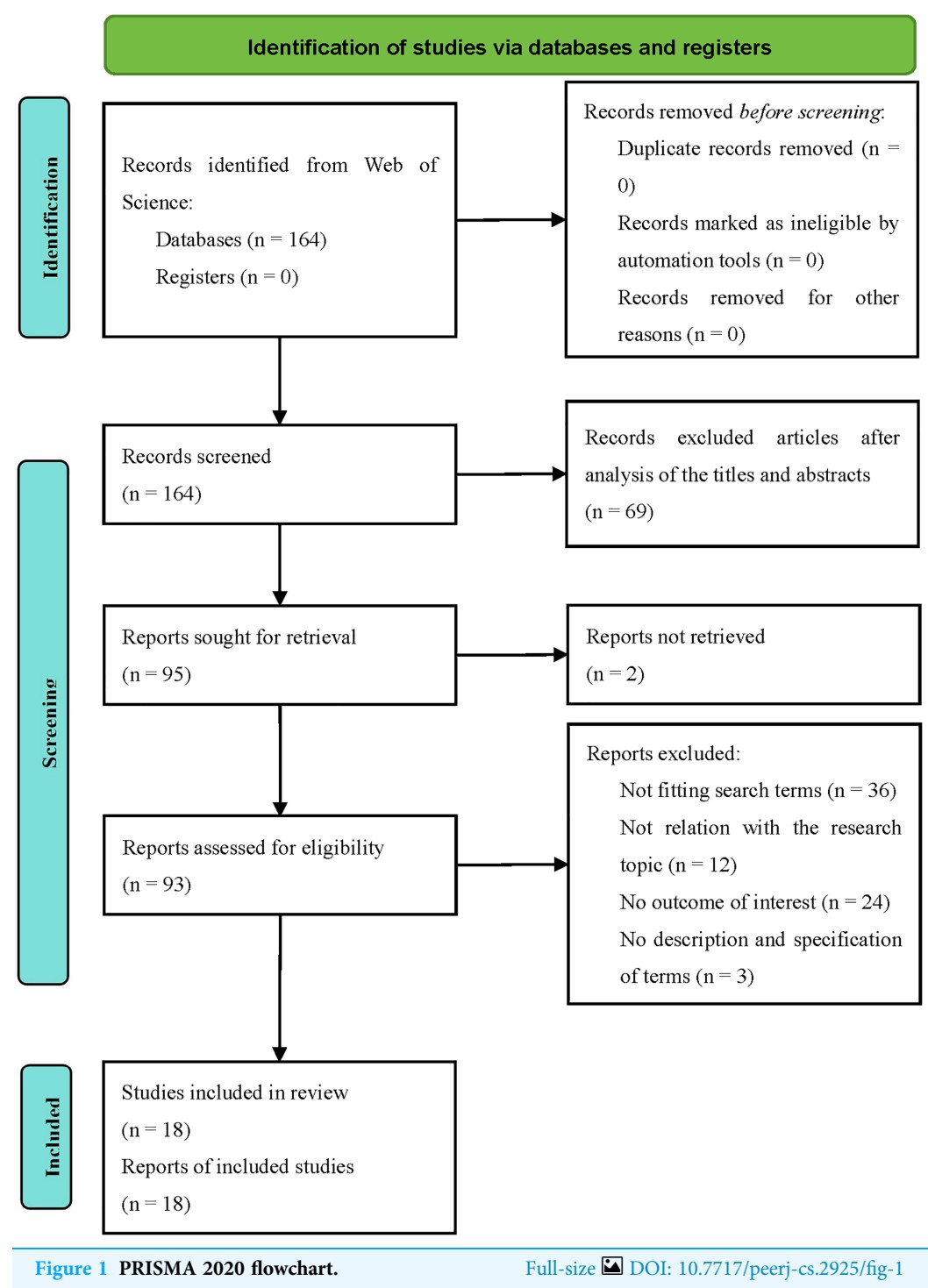

**Figure 1 PRISMA 2020 flowchart.**

The studies summarized in Table 1 indicate an increasing focus on applying blockchain technology to data trading, particularly regarding security and fairness. Among these studies, eight center on security (*Chen et al., 2019b*; *Camilo et al., 2020*; *Dai et al., 2019*; *An et al., 2022*; *Liu et al., 2022*; *Tian & Xiao, 2022*; *Zhao et al., 2023*; *Xu et al., 2024*), while four

**Table 1 Previous research on the application of blockchain in data trading.**

| Authors | Research description | Application direction |
|---|---|---|
| *Guan, Shao & Wan (2018)* | This study proposes two secure, fair, and efficient data trading schemes that do not rely on any third party using blockchain technology. The first scheme achieves direct raw data exchange of a large amount of data, while the second scheme implements data statistical trading. | Security & Fairness |
| *Chen et al. (2019b)* | This study proposes a secure and efficient blockchain based data trading scheme for connected vehicles. Apply consortium blockchain technology to ensure secure and authentic data trading. | Security |
| *Camilo et al. (2020)* | This study proposes a secure, agile, and effective system that utilizes blockchain, smart contracts, trust, and reputation to achieve distributed, automated, and transparent data trading between domains. The proposed system can provide security and privacy in a fast and distributed manner, execute hundreds of transactions per second, and effectively punish malicious behavior. | Security |
| *Dai et al. (2019)* | This study proposes a secure blockchain-based data trading ecosystem. In this ecosystem, both data brokers and buyers are unable to access the seller's raw data, reducing the challenge of ensuring dataset security to that of ensuring data processing security. | Security |
| *Li et al. (2020a)* | This study proposed a decentralized trading scheme for open and fair data trading by deploying smart contracts on a blockchain network. The data trading contract was implemented in a robust manner, achieving secure, practical, open, and fair trading. | Security & Fairness |
| *Li et al. (2020b)* | This study has developed a decentralized fair data trading framework, proposing two different potential instantiation models. In the first model, homomorphic encryption and data sampling techniques are utilized to enhance the system's reliability and further ensure the availability of data during the data trading process. In the second model, a signature preventing double authentication is integrated with smart contracts to achieve fairness in the data trading process. | Fairness |
| *An et al. (2022)* | This study proposes a blockchain based peer-to-peer secure data trading scheme for the Internet of Things. The Chameleon hash algorithm has been adopted to improve the robustness of trading schemes by detecting malicious data on the chain and implementing error correction and loss control. | Security |
| *Li, Li & Wang (2022)* | This study proposes a fair and decentralized data trading model based on blockchain technology. To ensure the fairness of data trading, sales contracts and deterministic public key encryption algorithms have been integrated. | Fairness |
| *Liu et al. (2022)* | This study proposes a new secure power data trading scheme. Zero-knowledge proofs are used to achieve the availability and consistency of data without revealing it. The decentralization and immutability of blockchain are fully utilized to ensure the reliability of data trading. | Security |
| *Tian & Xiao (2022)* | This study proposes a new blockchain-based copyright protection data trading system, which mainly consists of two blockchains. One of the blockchains is the copyright chain, which is designed to register and trade copyrights stored in atomic transaction form. The other is the usage rights blockchain, which records usage rights transactions and stores data. | Security |
| *An et al. (2021)* | This study proposes a blockchain-based crowdsensed data trading system. The blockchain is used to replace the data trading intermediary, ensuring the trustworthiness of data trading. The reverse auction mechanism based on blockchain is realistic and individual rational, which can ensure that data sellers report their data collection costs honestly and prevent sellers from manipulating the auction. | Security & Fairness |
| *Park, Jeon & Shin (2023)* | This study proposes a secure and fair data trading system based on blockchain technology. To ensure nonrepudiation and fairness among all parties involved in the trading, the proposed system utilizes smart contracts to ensure that all parties honestly execute the data trading protocol. | Security & Fairness |
| *Xue et al. (2023)* | This study proposes a fair and privacy-preserving data trading scheme based on blockchain technology. By combining data attribute based credentials, encryption, and zero-knowledge proof, fairness among trading participants can be achieved; By constructing a Merkle hash tree on the ciphertext of the data with a signature at its root node, fine-grained data transactions are ensured and identity privacy is protected. | Security & Fairness |
| *Zhao et al. (2023)* | This study proposes a blockchain-based multimedia data trading scheme without a third party. Smart contracts are used to complete copyright tracking and ensure the security and traceability of multimedia data trading. | Security |
| *Li et al. (2024b)* | This study proposed a fair and responsible trading scheme for educational multimedia data based on blockchain. To achieve fair trading, a smart contract with a reasonable pricing model was implemented. | Fairness |

(Continued)

| Table 1 (continued) | | |
| --- | --- | --- |
| **Authors** | **Research description** | **Application direction** |
| *Xu et al. (2024)* | This study proposes a blockchain-based vehicular crowdsensing security data trading system. To ensure the accuracy of sensing data while maintaining data privacy, a lightweight privacy-preserving truth discovery algorithm is combined with the blockchain-based data trading process. | Security |
| *Abla et al. (2024)* | This study proposes a blockchain-based fair data trading protocol. It integrates probabilistic methods and fully homomorphic encryption to achieve fairness, and allows online arbitration if improper trading behavior is detected. | Fairness |
| *Xiong et al. (2024)* | This study presents a one-to-many distributed data trading scheme based on blockchain. It enables data sellers to sell one piece of data to multiple buyers at the same time, ensuring data security, efficient access, fair trading, and revenue maximization. | Security & Fairness |

belong to the fairness category (*Li et al., 2020b*; *Li, Li & Wang, 2022*; *Li et al., 2024b*; *Abla et al., 2024*). Additionally, six studies (*Guan, Shao & Wan, 2018*; *Li et al., 2020a*; *An et al., 2021*; *Park, Jeon & Shin, 2023*; *Xue et al., 2023*; *Xiong et al., 2024*) were included in the analysis as they address both security and fairness. Thus, this study may be useful for the current research.

The application of blockchain to data trading is driven by two main factors. On one hand, blockchain is proven to enhance the security of data trading. For example, *Dai et al. (2019)* highlighted the significance of blockchain data processing as a service model. This model can effectively complement the traditional data hosting/swapping as a service model, thus alleviating many security limitations of traditional data trading platforms caused by dishonest buyers or data brokers. On the other hand, blockchain can also ensure fairness in data trading. For example, *Li et al. (2020b)* proposed a blockchain method for conducting data trading without the need for a centralized trusted third party. This method avoids the risk of data sellers sending data to buyers before receiving payment, which could lead to buyers obtaining data without payment. It also prevents data buyers from failing to receive data after payment.

The following section reviews the methodologies used in previous research on blockchain-based data trading. In addition, it examines the units of analysis that have been the focus of previous studies, as well as the scientific research areas that analyze and identify secure and fair data trading. The implications for managers and researchers are also analyzed.

## Methodologies used in previous research

The current systematic review literature predominantly centers on blockchain—based data trading. Within this field, the first key research direction is conceptual design solutions. Here, blockchain is utilized to tackle data trading security and fairness (*Guan, Shao & Wan, 2018*; *Chen et al., 2019b*; *Li et al., 2020a*; *An et al., 2022*; *Liu et al., 2022*; *Xue et al., 2023*; *Zhao et al., 2023*; *Li et al., 2024b*; *Xiong et al., 2024*).

The second direction is developing blockchain-based data trading systems. It focuses on software development to ensure data trading security and fairness. Different data types

**Table 2  Categorizes previous studies by methodology.**

| Authors | Scheme | System | Model |
|---|---|---|---|
| Guan, Shao & Wan (2018) | ✓ | | |
| Chen et al. (2019b) | ✓ | | |
| Li et al. (2020a) | ✓ | | |
| An et al. (2022) | ✓ | | |
| Liu et al. (2022) | ✓ | | |
| Xue et al. (2023) | ✓ | | |
| Zhao et al. (2023) | ✓ | | |
| Li et al. (2024b) | ✓ | | |
| Xiong et al. (2024) | ✓ | | |
| Camilo et al. (2020) | | ✓ | |
| Dai et al. (2019) | | ✓ | |
| Tian & Xiao (2022) | | ✓ | |
| An et al. (2021) | | ✓ | |
| Park, Jeon & Shin (2023) | | ✓ | |
| Xu et al. (2024) | | ✓ | |
| Li et al. (2020b) | | | ✓ |
| Li, Li & Wang (2022) | | | ✓ |
| Abla et al. (2024) | | | ✓ |

inspire distinct trading system designs (*Camilo et al., 2020*; *Dai et al., 2019*; *Tian & Xiao, 2022*; *An et al., 2021*; *Park, Jeon & Shin, 2023*; *Xu et al., 2024*).

The third direction involves blockchain-based data trading models. Researchers aim to find new rules to solve security and fairness issues and apply them to trading platform standards (*Li et al., 2020b*; *Li, Li & Wang, 2022*; *Abla et al., 2024*). Table 2 categorizes the reviewed studies into these three groups.

## Unit of analysis

The research reviewed in this article explores the diverse applications of blockchain technology in data trading. A significant portion of the studies (*Chen et al., 2019b*; *Camilo et al., 2020*; *Dai et al., 2019*; *An et al., 2022*; *Liu et al., 2022*; *Tian & Xiao, 2022*; *Zhao et al., 2023*; *Xu et al., 2024*) centers on the security of data trading, aiming to enhance the establishment of secure data trading frameworks using blockchain. Another set of studies (*Li et al., 2020b*; *Li, Li & Wang, 2022*; *Li et al., 2024b*; *Abla et al., 2024*) focuses on the fairness of data trading, proposing blockchain-based solutions to ensure equitable transactions. Additionally, several studies (*Guan, Shao & Wan, 2018*; *Li et al., 2020a*; *An et al., 2021*; *Park, Jeon & Shin, 2023*; *Xue et al., 2023*; *Xiong et al., 2024*) address both security and fairness, striving to develop comprehensive blockchain-based systems that ensure secure and fair data trading. These studies are categorized into the three aforementioned groups in Table 3.

**Table 3 Categorizes previous studies by unit of analysis.**

| Authors | Security | Fairness | Security & Fairness |
|---|---|---|---|
| *Chen et al. (2019b)* | ✓ | | |
| *Camilo et al. (2020)* | ✓ | | |
| *Dai et al. (2019)* | ✓ | | |
| *An et al. (2022)* | ✓ | | |
| *Liu et al. (2022)* | ✓ | | |
| *Tian & Xiao (2022)* | ✓ | | |
| *Zhao et al. (2023)* | ✓ | | |
| *Xu et al. (2024)* | ✓ | | |
| *Li et al. (2020b)* | | ✓ | |
| *Li, Li & Wang (2022)* | | ✓ | |
| *Li et al. (2024b)* | | ✓ | |
| *Abla et al. (2024)* | | ✓ | |
| *Guan, Shao & Wan (2018)* | | | ✓ |
| *Li et al. (2020a)* | | | ✓ |
| *An et al. (2021)* | | | ✓ |
| *Park, Jeon & Shin (2023)* | | | ✓ |
| *Xue et al. (2023)* | | | ✓ |
| *Xiong et al. (2024)* | | | ✓ |

## Scientometric analysis

To better understand which research fields prioritize data trading security and fairness, we conducted a scientometric analysis. Scientometric analysis, a quantitative and qualitative method for studying science and scientific achievements first proposed by *Price (1963)*, has been widely used as a supplementary analysis for systematic literature reviews (*Azarian et al., 2023*) or as a major research topic (*Di Zio et al., 2023*).

Table 4 summarizes the results of qualitative and quantitative analysis, including authors, Journal Citation Reports (JCR)-ranked journals and their categories, and quartiles. The author field is used to track the analysis conducted by the author throughout the entire article. The quartiles and categories reflect all the categories that the journal has according to the Web of Science classification, and if the quartiles of different categories are different, they will also be reflected in the table.

From Table 4, it can be seen that the discipline with the highest number of published articles is computer science. There are a total of 16 publications belonging to the computer science category. Next is engineering, with nine publications. Other disciplines include telecommunications (six articles), transportation, materials science, and mathematics (one article each). This indicates that both computer scientists and experts in the engineering field are interested in secure and fair data trading.

Additionally, it is worth noting that in the Web of Science database, there are four conference proceedings published articles from which we can extract categories, but we cannot perform a quartile analysis on them.

**Table 4 Scientometric classification.**

| Authors | Journal | Quartile | Category |
|---|---|---|---|
| *Guan, Shao & Wan (2018)* | 2018 IEEE International Conference on Internet of Things (iThings) | – | Computer Science; Engineering; Telecommunications |
| *Chen et al. (2019b)* | IEEE Transactions on Vehicular Technology | Q1 | Engineering; Telecommunications; Transportation |
| *Camilo et al. (2020)* | 3rd IEEE International Conference on Blockchain (Blockchain) | – | Computer Science |
| *Dai et al. (2019)* | IEEE Transactions on Information Forensics and Security | Q1 | Computer Science; Engineering |
| *Li et al. (2020a)* | Concurrency and Computation-Practice & Experience | Q2 | Computer Science |
| *Li et al. (2020b)* | IEEE Network | Q1 | Computer Science; Engineering; Telecommunications |
| *An et al. (2022)* | 4th ACM International Symposium on Blockchain and Secure Critical Infrastructure (BSCI) | – | Computer Science |
| *Li, Li & Wang (2022)* | Computer Journal | Q2 | Computer Science |
| *Liu et al. (2022)* | IEEE Transactions on Network and Service Management | Q1 | Computer Science |
| *Tian & Xiao (2022)* | 18th IEEE International Conference on Mobility, Sensing and Networking (MSN) | – | Computer Science; Engineering; Telecommunications |
| *An et al. (2021)* | IEEE Transactions on Mobile Computing | Q1 | Computer Science; Telecommunications |
| *Park, Jeon & Shin (2023)* | Cmc-Computers Materials & Continua | Q3 | Computer Science; Materials Science |
| *Xue et al. (2023)* | IEEE Transactions on Computers | Q2 | Computer Science; Engineering |
| *Zhao et al. (2023)* | Mathematics | Q1 | Mathematics |
| *Li et al. (2024b)* | Wireless Networks | Q3 | Computer Science; Engineering; Telecommunications |
| *Xu et al. (2024)* | ACM Transactions on Embedded Computing Systems | Q2 | Computer Science |
| *Abla et al. (2024)* | IEEE Transactions on Information Forensics and Security | Q1 | Computer Science; Engineering |
| *Xiong et al. (2024)* | IEEE Transactions on Information Forensics and Security | Q1 | Computer Science; Engineering |

As for the quartiles of the remaining articles, if there are multiple quartiles for different categories, we select the category with the highest quartile score. Therefore, eight articles are in Q1, four in Q2, and two in Q3.

Following this analysis, the implications for managers and researchers are then presented.

## Implications for managers

The results of this study highlight the importance of security and fairness in data trading, especially on data trading platforms, where platform managers need to pay attention and make appropriate technical adjustments and management (*Wang, Yuan & Li, 2023*).

In today's digital economy era, data is considered a valuable asset. This necessitates secure and fair data trading platforms (*Sestino, Kahlawi & De Mauro, 2023*). Platform managers must recognize that ensuring the security and fairness of data trading, leveraging blockchain technology's characteristics, can help maintain the platform's competitiveness in the industry (*Nuttah et al., 2023*).

The time and space inconsistencies between buyers, sellers, and data commodities in data trading lead to information asymmetry for data buyers regarding the trading

platform. However, when platform managers use blockchain technology, this information asymmetry is reduced (*Tian & Hu, 2023*).

As data's value gains attention, data trading shows promising market prospects, prompting many data trading platform managers to explore secure and fair trading and consider applying new technological strategies (*Gondo, Holtman & Rena, 2023*).

Blockchain technology can provide decentralized, tamper-proof, traceable, and transparent data trading for platforms (*Mohialden & Hussien, 2024*). Therefore, it is crucial to strengthen the construction of blockchain-based data trading, avoid unsafe and unfair trading, and develop blockchain systems to support and promote secure and fair data trading.

### Implications for researchers

Despite extensive research on secure and fair data trading and proposed solutions, there is still room for improvement on data trading platforms. Researchers in this field should have a comprehensive understanding of existing research results.

Previous studies on data trading security and fairness have primarily employed blockchain technology, including blockchain-based data trading schemes (*Guan, Shao & Wan, 2018*; *Chen et al., 2019b*; *Li et al., 2020a*; *An et al., 2022*; *Liu et al., 2022*; *Xue et al., 2023*; *Zhao et al., 2023*; *Li et al., 2024b*; *Xiong et al., 2024*), systems (*Camilo et al., 2020*; *Dai et al., 2019*; *Tian & Xiao, 2022*; *An et al., 2021*; *Park, Jeon & Shin, 2023*; *Xu et al., 2024*), and models (*Li et al., 2020b*; *Li, Li & Wang, 2022*; *Abla et al., 2024*). Researchers can strengthen their academic theoretical framework by applying our research results through the correct selection of research methods.

For future research, the selection of the units of analysis is of critical importance, as it determines both the granularity and applicability of findings. Limited prior work has focused on trading fairness (*Li et al., 2020b*; *Li, Li & Wang, 2022*; *Li et al., 2024b*; *Abla et al., 2024*). In contrast, most studies have prioritized trading security (*Chen et al., 2019b*; *Camilo et al., 2020*; *Dai et al., 2019*; *An et al., 2022*; *Liu et al., 2022*; *Tian & Xiao, 2022*; *Zhao et al., 2023*; *Xu et al., 2024*). Beyond this dichotomy, some studies additionally examine the security and fairness in data trading (*Guan, Shao & Wan, 2018*; *Li et al., 2020a*; *An et al., 2021*; *Park, Jeon & Shin, 2023*; *Xue et al., 2023*; *Xiong et al., 2024*).

## LIMITATIONS AND DISCUSSIONS

From an overall perspective, most existing representative studies have focused on leveraging blockchain technology to address the critical issues of security and fairness in data trading. Blockchain's decentralization and immutability enhance transaction security and data authenticity, while fostering fairer trading processes compared to traditional centralized systems, which often face challenges such as manipulation and data tampering that compromise fairness. However, despite these advantages, existing research has paid insufficient attention to the efficiency challenges arising from blockchain technology use.

These efficiency limitations are actually a direct consequence of the fundamental principles of decentralization and fairness. Decentralization, which lies at the heart of

blockchain systems, necessitates consensus mechanisms such as Proof of Work (PoW) or Proof of Stake (PoS) to validate transactions across a vast number of decentralized nodes. While these mechanisms are key to ensuring security and fairness, for example, Bitcoin's PoW algorithm, which effectively prevents malicious entities from manipulating transactions, inevitably leads to a substantial loss in efficiency. The fact that Bitcoin can process only seven transactions per second, and Ethereum just a few dozen transactions per second, clearly shows that building decentralized consensus naturally restricts scalability. Thus, when striving for decentralization and fairness, compromises in transaction-processing efficiency are often required, and this is a major flaw that existing research hasn't fully and effectively addressed.

Efficiency is crucial for blockchain technology's application across economic and social domains. Bitcoin enhances efficiency through sidechains and the Lightning Network, though these solutions compromise fairness by centralizing small transactions. Ethereum seeks improvement *via* the Proof-of-Stake (PoS) algorithm and sharding technology, yet sharding reduces decentralization by dividing nodes. EOS achieves over 2,000 transactions per second (TPS) using the Delegated Proof-of-Stake (DPoS) mechanism with 21 super nodes, trading decentralization for efficiency.

Security is paramount in blockchain applications, especially given their association with digital currencies and assets. Despite innovations, security concerns persist, as vulnerabilities can lead to irreparable losses. All blockchain systems, from decentralized Bitcoin to multi-centered EOS, must address security challenges.

To effectively apply blockchain technology to data trading, three key issues require urgent attention:

(1) Balancing efficiency and security: blockchain-based data trading systems need both efficiency and security, yet achieving both simultaneously is difficult. Bitcoin's PoW offers high, mathematically proven security but only 7 TPS. Lowering PoW difficulty to boost consensus speed undermines system security. Existing solutions replacing PoW with PoS, DPoS, or Proof of Authority (PoA) lack proven security.

(2) Balancing transparency and privacy: in blockchain-based data trading, transparency is valued for fair transactions, but it conflicts with privacy. Consortium chains, which balance transparency and privacy, are one solution, yet public chain scenarios still struggle to achieve this balance.

(3) Balancing high concurrency and high throughput: data trading systems require high concurrency to handle numerous simultaneous users and trading requests. However, current blockchain solutions fall short in transaction throughput, failing to meet system requirements. Traditional blockchains use synchronous consensus, verifying transactions before adding them to the chain to synchronize node states. Asynchronous consensus solutions like Tangle, IOTA, and XDAG (using a directed acyclic graph structure) aim to balance concurrency and throughput by first attaching then verifying transactions, though their effectiveness remains unproven.

## CONCLUSIONS

This systematic review synthesizes the evolving landscape of blockchain applications in data trading, yielding three pivotal insights.

First, blockchain has emerged as a transformative framework for ensuring secure and fair data trading, gaining significant interdisciplinary traction in computer science and engineering. Recent studies (*Liu et al., 2022*; *An et al., 2021*; *Xiong et al., 2024*) show its growing use in addressing vulnerabilities in conventional platforms. It does this particularly through decentralizing trust and automating compliance *via* smart contracts. This trend aligns with the need to reconcile user reliance on imperfect trading systems (*Andrus et al., 2021*; *Tian et al., 2024*) with robust technological safeguards.

Second, methodological analysis reveals a strong focus on computer science-driven innovations. Of the 18 studies reviewed, 83% center on schemes (nine studies) and systems (six studies), emphasizing algorithmic improvements and architectural redesigns over theoretical modeling (three studies). This dominance underscores the field's applied nature, where solutions like zero-knowledge proofs (*Xue et al., 2023*; *Abla et al., 2024*) and hybrid consensus mechanisms (*Xu et al., 2024*) are tested to balance security and efficiency.

Third, our findings pinpoint three critical dimensions for advancing blockchain-based platforms:

(1) Efficiency-security equilibrium: requires lightweight cryptography to cut computational overhead.
(2) Transparency-privacy duality: needs adaptive frameworks, such as blockchains integrated with federated learning.
(3) Scalability: demands sharding or layer-2 protocols to boost throughput without sacrificing decentralization.

While this review uses the Web of Science *corpus*, expanding to domain-specific databases (*e.g.*, IEEE Xplore, arXiv) could reduce potential publication bias. Future work should focus on two areas:

(1) Quantitative impact assessments: use metrics like transaction finality time or attack resistance rates to benchmark blockchain against traditional systems.
(2) Multi-objective optimization: identify trade-offs among fairness, security, and efficiency using models like game theory or reinforcement learning. Also, legal-technical interdisciplinary studies are crucial to establish user recourse mechanisms in adversarial situations.

In conclusion, blockchain's potential to recalibrate data trading ecosystems depends on moving beyond technical proofs-of-concept to holistic, regulation-aware architectures. This shift requires joint efforts to balance cryptographic rigor with usability, ensuring platforms resist exploitation and foster sustainable trust among stakeholders.

### Funding

This work was supported by the Social Science Foundation Project of Jiangxi Province (No. 22GL28), the Science and Technology Research Project of Education Department of Jiangxi Province (No. GJJ2201507), the Key Project of Natural Science Foundation of Jiangxi Province (No. 20232ACB201003), and the National Natural Science Foundation of China (No. 72271113). The funders had no role in study design, data collection and analysis, decision to publish, or preparation of the manuscript.

### Grant Disclosures

The following grant information was disclosed by the authors:
Social Science Foundation Project of Jiangxi Province: 22GL28.
Science and Technology Research Project of Education Department of Jiangxi Province: GJJ2201507.
Key Project of Natural Science Foundation of Jiangxi Province: 20232ACB201003.
National Natural Science Foundation of China: 72271113.

### Competing Interests

The authors declare that they have no competing interests. Wei Xiong is employed part-time as a researcher by Aheadsoft Software Co., Ltd.

### Author Contributions

- Wei Xiong conceived and designed the experiments, performed the computation work, prepared figures and/or tables, authored or reviewed drafts of the article, and approved the final draft.
- Huaibin Shao performed the experiments, authored or reviewed drafts of the article, and approved the final draft.
- Hong Ge analyzed the data, prepared figures and/or tables, and approved the final draft.

### Data Availability

This is a literature review.

### Supplemental Information

Supplemental information for this article can be found online at http://dx.doi.org/10.7717/peerj-cs.2925#supplemental-information.

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
