# Peer review of "The application of blockchain technology in data trading: a systematic review"

_PeerJ Computer Science, doi:10.7717/peerj-cs.2925_

## Round 0.1 · original submission · Major Revisions

The paper needs a lot of improvement. Specifically the reviewer 2 comments.

Reviewer 1 ·

Basic reporting

The paper provides a comprehensive overview of the application of blockchain technology in the context of big data trading. It delves into various aspects of blockchain-based systems, including consensus mechanisms, data storage methods, and dispute arbitration. The paper aims to address the critical issues of security and fairness in big data trading, offering a comparative analysis of traditional and blockchain-based trading models.

Overall, the paper needs improvements on several aspects, as explained below:

1. The paper primarily focuses on theoretical and literature-based analysis without offering empirical data to validate the proposed benefits of blockchain technology in big data trading. While the paper discusses the potential advantages of blockchain, it lacks concrete examples or case studies that demonstrate the real-world effectiveness of these solutions.

2. Although the paper outlines the benefits of blockchain technology, it does not adequately address the challenges and limitations associated with its implementation. Issues such as scalability, energy consumption, and the complexity of integrating blockchain with existing systems are not sufficiently explored.

3. Many pertinent studies are still missing in the literature review of this article

Experimental design

4. The paper's methodology section is somewhat lacking in detail. While it mentions the systematic review of existing literature, it does not provide a clear explanation of the criteria used for selecting the studies or the process of analyzing them. This makes it difficult to assess the rigor of the review process.

5. The paper contains some redundancy, particularly in the sections discussing the advantages of blockchain technology. The same points are repeated in different sections, which could be streamlined for clarity and conciseness.

Validity of the findings

6. The paper briefly mentions areas for future research but does not provide a detailed roadmap or specific questions that need to be addressed. A more thorough exploration of future directions would enhance the paper's contribution to the field.

7. While the focus on security and fairness is commendable, the paper could benefit from a broader discussion of other critical factors in big data trading, such as efficiency, cost, and user adoption. A more balanced analysis would provide a more comprehensive view of the potential of blockchain technology in this context.

Cite this review as

Reviewer 2 ·

Basic reporting

1. The abstract of the paper lacks clarity and depth. The authors failed to present the complete idea of the paper in the abstract. I recommend completely rewriting the entire abstract.

2. There is no flow in the starting paragraph of the introduction section. I strongly recommend maintaining consistency and coherence between each sentence.

3. In the introduction section, the sentence " Big data trading is conducive to mining the potential value of data resources, and is conducive to playing the role of data as important elements as land and energy, and is conducive to promoting data flow to lead material flow, capital flow, talent flow, and technology flow, and is conducive to promoting industrial model innovation and industrial transformation and upgrading." is very complex and unclear. Please avoid complex and meaningless sentences in any part of the paper.

4. The authors have not discussed the research questions. The authors should include some research questions that provide motivation for this survey paper. The authors should also provide all possible solutions for sorting out the research questions.

5. A good review/survey paper has a proper taxonomy diagram. A proper diagram about the taxonomy of the paper is missing. I strongly recommend adding a proper taxonomy diagram to explain the complete layout of the paper.

6. The authors have not discussed the criteria for paper selection. It is necessary to explain how the papers in the literature were selected and how the gaps were identified.

7. Big data trading modes are not properly defined and described. The authors are suggested to include more literature in these sections. I strongly recommend adding separate tables in each section to describe the previous research studies conducted by numerous research scholars.

8. Every main section should have a table for presenting the earlier studies by adding columns with the following names: S. No, Reference Article, Methodology, Outcomes/Results, Contributions, and Limitations.

9. Very poor representation of literature.

10. The grammar of the paper needs significant improvements.

11. All the figures listed in the paper need improvement.

12. All the tables listed in the paper are quite simple and ordinary. I recommend revising them.

13. The authors have compared the paper with only one paper. The authors should compare the paper with at least 10 papers.

14. Conclusion of the paper is weak. The current challenges and future directions are not properly described.

15. The paper has so many technical and grammatical issues. I recommend a thorough proofreading of the entire paper and addressing all of the technical and grammatical issues.

Experimental design

The paper needs significant improvement. I am not satisfied with the study design of this paper.

Validity of the findings

The findings of the paper are not convincing.

Additional comments

I recommend complete revision of the paper.

Cite this review as

Reviewer 3 ·

Basic reporting

This paper offers a valuable contribution to the understanding of blockchain-based big data trading. The focus on decentralized trading mechanisms aligns well with current trends in data management and security. However, some revisions would be beneficial to further enhance the paper’s clarity and impact.
1: Comparing it with more recent studies could enhance its relevance and highlight how it stands out from other contemporary research. Including recent advancements in blockchain would also provide a more up-to-date perspective.
2: Comparing the differences between this work and other research in the area of big data trading positions the paper as a significant contribution. However, it would be beneficial to highlight the unique contributions of this paper more explicitly.

Experimental design

3: The paper provides a qualitative comparison of Proof of Work (PoW) and Proof of Authority (PoA), which is valuable. However, it would be more comprehensive if the comparison were extended to include other consensus mechanisms such as Proof of Stake (PoS) and Delegated Proof of Stake (DPoS) and others. Including these additional mechanisms would offer a broader perspective and help readers better understand the strengths and weaknesses of each approach in the context of big data trading.
4: The figures in the paper are not explained logically and lack comprehensive detail. To enhance the clarity and effectiveness of the visual aids, each figure should be accompanied by a thorough explanation that connects it to the main text. This will help readers better understand the data being presented and how it supports the paper's arguments.

Validity of the findings

No comment

Additional comments

5: The overall structure of the paper could benefit from improvements in logical flow and organization. Some sections feel disjointed, and the connections between different parts of the paper are not always clear. To strengthen the paper, it would be helpful to revise the structure to ensure a more coherent progression of ideas, with clear transitions between sections.
6: The paper would benefit from including a brief overview of blockchain in the context of big data trading. This should cover key concepts like decentralization, security, transparency, and smart contracts, along with the advantages and challenges. Adding this context would enhance the paper's comprehensiveness.

Cite this review as

---

## Round 0.2 · Major Revisions

The paper needs to be improved further, according to the reviewers suggestions

Reviewer 1 ·

Basic reporting

The authors have carfully addressed the reviewers' comments; no futrther revision is required.

Experimental design

The study design has significantly been improved.

Validity of the findings

This study aims to deeply explore data trading
based on blockchain. To achieve this, we conducted a systematic review of relevant
literature, focusing on two main themes: (i) data trading and (ii) blockchain. We collected
relevant literature using scientiûc databases and used "data trading" and "blockchain" as
search keywords, and a total of 162 articles were retrieved from the Web of Science
database. After applying diûerent ûlters according to the PRISMA 2020 ûowchart, we
ûnally selected 16 studies. The research results show that: (i) the analysis of the
application of blockchain in data trading is mainly concentrated in the ûeld of computer
science; (ii) the application of blockchain technology is mainly based on solutions and
systems, and model methods are applied relatively less; (iii) decentralized data trading
based on blockchain faces multiple challenges in terms of eûciency and security,
transparency and privacy, high concurrency and high throughput in both the present and
the future.

Cite this review as

Reviewer 3 ·

Basic reporting

The authors have addressed the comments thoroughly; however, there is still a need to improve the paper’s fluency and structure to enhance readability and coherence. Further refinement of the language and organization would make the arguments clearer and strengthen the overall quality of the manuscript.

Experimental design

no comment

Validity of the findings

no comment

Additional comments

no comment

Cite this review as

·

Basic reporting

Comment 1
There are some grammatical error that makes reading difficult. The manuscript in general should be read and corrected.

Comment 2
In many instances there is still lack of continuity and consistent flow from one sentence to the other. It disrupts reading flow and clarity. For example, it is not clear the intention of lines 31-39 with each sentence entirely different from the previous. Lines 69-72 should be re-written to improve clarity.

Experimental design

Comment 1
Clear conclusions from the review are still missing from the abstract and conclusions. On lines 91-92 the reason for the review of current direction of research is not stated. The research questions being answered is still missing.

Comment 2
The introduction is unclear about the purpose of the review. It has failed to highlight clearly and concisely the current challenges (security and efficiency) with existing system necessitating the need for the study.

Comment 3
Lines 79-81 needs more referencing and clarity especially because low-cost, high efficiency, high accuracy are not in themselves inherent in the technology and can vary across selected blockchain platform.

Comment 4
Clearer overview of blockchain context and key terminologies are still inadequate for a review paper targeted at a wider audience.

Comment 5
The manuscript is still missing recent advancements and clear conclusions or limitations from previous review works in the literature review. This could help justify the goal of the review and distinguish it from others.

Comment 6
In the Methodology, the selection process is not well presented. The results of a systematic review process should ideally be reproducable if the steps are clearly defined.
The authors should justify why randomized control process was preferred.
On line 140-141 it is not clear if only Web of Science was the only database used. If only, this should also be justified as this could be potential limitation.
On line 147, kindly justify high-quality research.
The use of PRISMA should be justified and the concept briefly presented.
On line 149-150, the statement contradicts the actual action taken in line 141-142 in the sense that it seems that the search was done against only one database.
There is a discrepancy between 162 papers mentioned in the abstract and 152 presented in the methodology.
The reduction from 152 to 90 is not clear. What are the concise exclusion criteria for eliminating 62 papers. The numbers of papers eliminated for each criteria should be stated.
The data extraction criteria should be presented earlier before being used in line 173-175.
The years concerned should stated and extended. 16 papers may not be a fair representation of many previous works on data trading using blockchain

Comment 7
The Wider Perspective section is not part of the conclusions from the review and such come earlier in the Introduction to provide better context to blockchain and consensus mechanisms.
Meanwhile, only PoW consensus was mentioned. Nowadays, there are many other consensus with better TPS more widely employed for data trading. This should be explored and stated.

Validity of the findings

Comment 1
There are no concrete synthesized information from the analysis that distinguishes the review from previous reviews. More in-depth analysis is required.

Comment 2
There are no clear conclusions from the analysis.

Additional comments

The authors made attempts to review previous works employing blockchain for data trading. The approach using systematic review is a good one. However, in general there is lack of clarity in the review process, absence of detailed analysis and concrete conclusions. Many of the comments from previous reviewers are still not properly addressed. The work requires significant improvements.

---

## Round 0.3 · Minor Revisions

The authors improved the paper. However, one reviewer still recommended some changes to do.

Reviewer 2 ·

Basic reporting

The authors have improved the article.

Experimental design

It looks fine, but it can be improved by considering multiple parameters.

Validity of the findings

It looks fine, but can be improved.

Additional comments

The authors have improved the paper, but it can be improved by considering multiple parameters for evaluation. Also, considering recent research in this domain will further enhance the worth of this article.

Cite this review as

·

Basic reporting

The authors have significantly and adequately resolved grammatical errors. They have also greatly and commendably improved the flow in reading.

Experimental design

The approach is generally sufficiently described. The following comments will further improvement it.

Comment 1:
Kindly state clearly the years of previous works covered by the review.

Comment 2:
Line 157 jumps into describing PRISMA without presenting the context – A statement mentioning that a systematic review process following PRISMA was employed.

Comment 3:
Lines 135-176 should be re-written to improve clarity. First introduce the goal, describe the tools, state the date of the collection, explain the collection and elimination process with numbers breakdown and justifications.

Validity of the findings

The conclusion looks adequate and sufficient, but the analysis needs some improvements. It is quite difficult to deduce the conclusions from the analysis. I have suggested the following improvements.

Comment 1:
The “Exploratory Analysis of Results” still requires significant improvements. Please you values such as “5 of the studies applied …”. It helps to concretize the analysis.

Comment 2

The following statement is ambiguous - “The following section reviews the methodology used in previous research on blockchain-based data trading, focusing particularly on the units of analysis …” - is the unit of analysis a methodology used in previous research?
If possible, improve the discussions in the sections of Unit of Analysis and the Scientometric Analysis.

Comment 3
The “Implications” section should be part of the Analysis of Result and not as a separate section. Improve the discussion here to clearly show the value to Platform managers in taking decisions? and to researchers in the direction for future research.

Comment 4
Why is this so “For future research, it is important to consider analytical units. While some studies focus on trading fairness” – how does it affect research works?

Comment 5
In the “Limitations and Discussion” you discussed about efficiency but in your Result Analysis, there is no discussion or analysis on efficiency. Add to the result analysis section discussions about efficiency deduced from the review to support claims.

---

## Round 0.4 · accepted · Accept

The paper has been improved.

·

Basic reporting

Comments have been addressed.

Experimental design

No comment

Validity of the findings

No comment